# Multi-way Variational NMT for UGC: Improving Robustness in Zero-shot Scenarios via Mixture Density Networks

**José Carlos Rosales Núñez**
Université Paris Saclay & LISN
INRIA Paris
jose.rosales-nunez@inria.fr

**Djamé Seddah**
INRIA Paris
djame.seddah@inria.fr

**Guillaume Wisniewski**
Université Paris Cité,
LLF, CNRS
guillaume.wisniewski@u-paris.fr

## Abstract

This work presents a novel Variational Neural Machine Translation (VNMT) architecture with enhanced robustness properties, which we investigate through a detailed case-study addressing noisy French user-generated content (UGC) translation to English. We show that the proposed model, with results comparable or superior to state-of-the-art VNMT, improves performance over UGC translation in a zero-shot evaluation scenario while keeping optimal translation scores on in-domain test sets. We elaborate on such results by visualizing and explaining how neural learning representations behave when processing UGC noise. In addition, we show that VNMT enforces robustness to the learned embeddings, which can be later used for robust transfer learning approaches.

## 1 Introduction

The specificities of user-generated content (UGC) leads to a wide range of vocabulary and grammar variations (Foster, 2010; Seddah et al., 2012; Eisenstein, 2013). These variations result in a large increase of out-of-vocabulary words (OOVs) in UGC corpora with respect to canonical parallel training data and raise many challenges for Machine Translation (MT), all the more since common language variations found in UGC are actually productive (there will always be new forms that will not have been seen during training). This fact limits the pertinence of "standard" domain adaptation methods such as fine-tuning[1] or normalization techniques (Martínez Alonso et al., 2016) and urges the development of robust machine translation models able to cope with out-of-distribution (OOD) texts in a challenging zero-shot scenario in which the target distribution is unknown during training.

To address the problem raised by OOD texts, an increasing number of works (Setiawan et al., 2020; Schmunk et al., 2013; McCarthy et al., 2020; Przystupa, 2020; Xiao et al., 2020) explore the possibility to combine deep learning with latent variable (LV) models, which are indeed able to capture underlying structure information and to model unobserved phenomena. The combination of these models with neural networks was shown to increase performance in several NLP tasks (Kim et al., 2018). In this work, we focus on a specific latent variable model for MT, Variational NMT (VNMT) (Zhang et al., 2016) which has been reported to have good performance and interesting adaptability properties (Przystupa, 2020; Xiao et al., 2020).

The goal of this work is twofold. First, we aim to evaluate the performance of VNMT when translating a special kind of OOD texts: French social-media noisy UGC. To account for the challenges raised by the productive nature of UGC, we consider a highly challenging zero-shot scenario and assume that only canonical texts[2] are available for training the system. We hypothesize that, by leveraging on Variational NMT, latent models can build more robust representations able to represent OOD observations that are symptomatic of noisy UGC and automatically map them to in-distribution instances, which can be more easily translated.

Furthermore, to account for the diversity of UGC phenomena, we introduce a new extension of VNMT that relies on Mixture Density Networks (Bishop, 1994) and Normalizing Flows (Rezende and Mohamed, 2015). Intuitively,

---

[1] As the fine-tuning data will only reflect a frozen state of idiosyncrasies.

[2] We consider the corpora generally used to train MT systems as "canonical" as they contain texts following the set of standard grammatical and morphological source-language rules.

each mixture component extracts an independent latent space to represent the source sentence and can potentially model different UGC specificities. Interestingly, extracting embeddings from our zero-shot model that has never seen any UGC data and using them in a classic transformer-based NMT model leads to a stronger, more robust to UGC noise model. This is in line with the regularizing character of VNMT (Zhang et al., 2016).

Our contributions can be summarized as follows:

- we study the performance, in a zero-shot scenario, of VNMT models and evaluate their capacity to translate French UGC into English, which resulted in a consistent improvement of translation quality;

- we introduce a new model that uses state-of-the-art transformer as the backbone of a variational inference network to produce robust representation of noisy source sentences, and whose results outperform strong VNMT and non-latent baselines when translating UGC in a zero-shot scenario. Specifically, our model demonstrates a high robustness to noise while not impacting in-domain translation performance;

- by probing the learned latent representations, we show the importance of using several latent distributions to model UGC and the positive impact of the ability of VNMT models to discriminate between noisy and regular sentences while maintaining their representation closer in the embedding space;

- we report evidence that our VNMT models act as regularizers of their backbone models, leading to more robust source embeddings that can be later transferred with a relatively high performance gain in our zero-shot UCG translation scenario.

## 2 Background and related works

**Variational Neural Machine Translation** Variational Inference (VI) methods (Kingma and Ba, 2015) are generative architectures capable, from a distributional perspective, of modeling the hidden structures that can be found in a corpus. In a sequence-to-sequence MT task, where $x$ and $y$ are respectively the source and target sentences, VNMT (Zhang et al., 2016) architectures assume there exists an hidden variable $z$ modeling the implicit structure (i.e. relations) between the bilingual sentence pairs. In the context of UGC translation, we believe that this latent variable can capture the variations between a source sentence and its canonical, normalized form, recovering its underlying meaning and ensuring that the representation of the former is close to the representation of the latter.

To make computations tractable, in spite of the latent variable, VI combines a so-called *variational posterior* $q_\phi(z|x, y)$ that is chosen to approximate the *true* posterior distribution, with prior $p(z|x)$; and a neural decoder generative distribution, $p_\theta(y|x, z)$, in charge of generating the translation hypothesis conditioned on the latent variable. Once the family of densities $q$ is chosen, the parameters of the two distributions are jointly estimated to model the output $y$ by looking for the parameters $(\theta, \phi)$ that maximizes the *evidence lower bound* objective function:

$$
\begin{aligned}
\log p_\theta(y) \geq\ & \mathbb{E}_{q_\phi(z|x,y)}[\log p_\theta(y|x, z)] \\
& - D_{KL}[q_\phi(z|x, y)||p(z|x)]
\end{aligned}
\tag{1}
$$

**Normalizing Flows** One of the major caveats of variational methods is that choosing the prior $q(z)$ is a complicated process that requires some *a priori* knowledge of the task. In practice, a normal distribution with fixed parameters (generally $\mu = 0.0$ and $\sigma = 1.0$) is often chosen due to the simplicity of its re-parametrization for sampling. However, such an assumption can be restrictive when modeling more complex processes.

Regarding this issue, Rezende and Mohamed (2015) propose to enhance variational methods with Normalizing Flows (NF) (Tabak and Turner, 2013). A chain of normalizing flows is a series of simple bijective functions automatically chosen to extract a more suitable representation for the task at hand from a random variable, by alleviating the restrictions of choosing a default fixed prior. Concretely, a base distribution $q_0(z_0)$, that generates the initial latent codes, $z_0$, undergoes a series of invertible and smooth transformations $f : \mathbb{R}^d \to \mathbb{R}^d$, called *flows*. Then, the random latent variables $z$ are transformed to the random variable $z' = f(z)$ after each flow:

$$
q(z') = q(z)\left|det\frac{\partial f^{-1}}{\partial z'}\right| = q(z)\left|det\frac{\partial f}{\partial z}\right|^{-1}
\tag{2}
$$

Finally, we can build an arbitrarily $K$-long chain of $f_k$ transformations to generate the final latent variables, $z_K$, from the initial random variables, $z_0$, which is drawn from the base distribution $q_0(z_0)$ (often chosen to be $\mathcal{N}(0,1)$):

$$z_K = f_K \circ ... \circ f_2 \circ f_1(z_0)$$

$$ln(q_K(z_K) = ln(q_0(z_0)) - \sum_{k=1}^{K} ln \left| det \frac{\partial f_k}{\partial z_{k-1}} \right|$$

(3)

In MT, normalizing flows were recently used to improve VNMT models: Setiawan et al. (2020) show that using them in an in-domain evaluation setting results in an increase of +1.3 BLEU points on the IWSLT'14 (De-En) and +0.2 BLEU points on the WMT'18 (En-De); in a *simulated* out-domain evaluation, NF still improve translation quality: adding NF to the model trained on WMT'18 result in a +0.9 BLEU score improvements than the baseline Transformer system and +0.6 compared to the VNMT without using NF.

**Mixture Density Networks** Mixture Density Networks (MDN) are another interesting generalization of variational encoding for modeling UGC. By using MDN, the posterior distribution of the current decoding step $p(z|x, y_t)$ is no longer approximated by a single variational distribution $q_\phi(z|x, y_{1:t-1})$ but by a linear combination of variational posteriors $\tilde{q}_\phi^m(z|x, y_{1:t-1})$:

$$p(z|x, y_t) = \sum_{m=1}^{M} \alpha_m(x, y_{1:t-1}) \cdot \tilde{q}_m(z|x, y_{1:t-1})$$

(4)

where $\alpha_m$ are known as the mixing coefficients. Intuitively, an MDN can be interpreted as a combination of $M$ variational encoders. Our intuition is that, since UGC contains a large number of different kind of variations, covering very different aspects ranging from morphology to phonetics, including lexicon and sentence structure (Seddah et al., 2012); by using several independent VI components we can account for multiple UGC phenomena. Thus, with an MDN, it is possible that each component of the variational encoder is able to model different UGC specificities, allowing us to better process UGC as a whole. In the past, MDN has been used to address sequence-to-sequence generative tasks, such as SketchRNN (Ha and Eck, 2018) and modeling of sequential environment states in reinforcement learning (Ha and Schmidhuber, 2018).

**Gumbel-Softmax sampling** Regarding the mixing coefficients definition, we also explore the use of a categorical probability distribution, for which probabilities are calculated by the network, such as in Ha and Eck (2018). Unlike theirs, our supervised end-to-end training requires backpropagating the error gradient through the variational network via reparametrized sampling (Kingma and Welling, 2014) which poses optimization challenges because of the discrete random variables used as latent vector for categorical distributions. For this reason, we use the reparametrization of this distribution via the Gumbel-Softmax sampling (Jang et al., 2017; Maddison et al., 2017), such that, the $\arg\max$ function is approximated by a $\mathrm{softmax}$ and generates the relaxed one-hot encoded samples, which correspond to the mixing coefficients:

$$\alpha_m = \frac{\exp(\log(\pi_m) + g_m)/\tau)}{\sum_{j=1}^{M} \exp((\log(\pi_j) + g_j)/\tau)}$$

(5)

where $g_m...g_M$ are *i.i.d* samples from the Gumbel(0,1) distribution (Gumbel, 1954; Maddison et al., 2017), $\pi_i$ the probability associated to the $m$-th MDN's gaussian components, jointly generated by neural networks along with the computations of the corresponding parameters $(\mu_m, \sigma_m)$ for $m...M$; and $\tau$ the temperature parameter, which controls variability of the sampling. When $\tau \to 0$, the sampling exhibits a perfectly one-hot encoded output, whereas, conversely, when $\tau \to \inf$, the distribution approaches an uniform one across all the MDN's components.

## 3 Extending Variational Methods for Robust MT

Our model adopts a variational encoder-decoder architecture inspired by SketchRNN (§2) that uses an MDN on the decoder's variational network to model multiple and independent continuous generative variational distributions. However, unlike SketchRNN, we use a Transformer backbone for the encoder and the decoder and train our model in a end-to-end manner on canonical parallel corpora. In the following, we will first describe the general architecture of our model, denoted multi-VNMT, and then detail the encoder and decoder parameters.

**General architecture** Figure 1a represents the architecture of our model. The input sentence is

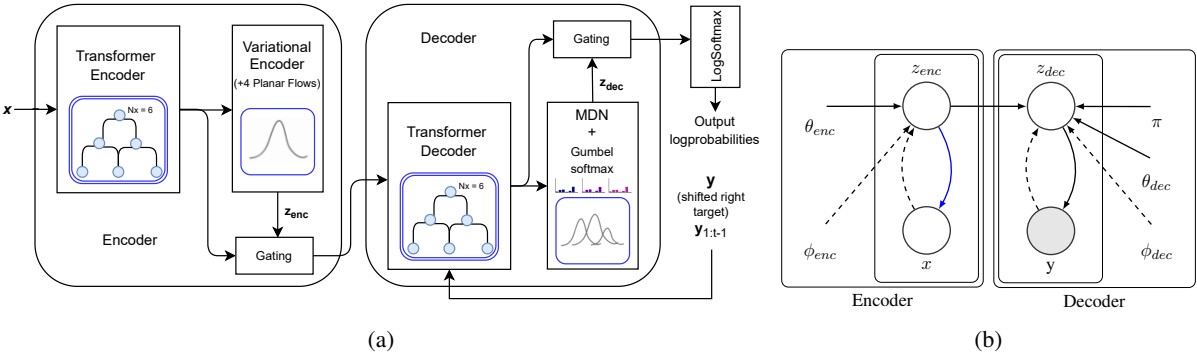

Figure 1: (a) `VNMT-MDN` architecture overview. (b) Directed graph of our encoder-decoder model variational inference. Dashed lines represent the variational approximation for the posterior distribution, and solid lines stand for the generative models. The blue arrow depicts the generative networks for source-side monolingual reconstruction distribution $p(\boldsymbol{x}|\boldsymbol{z})$.

first processed by a standard Transformer encoder, the output of which is used by a Variational Encoder enhanced with NF to predict a latent representation of the input sentence. The latent representation and the output of the last layer of the Transformer encoder are combined using the gating mechanism of Setiawan et al. (2020).

This combined representation is then fed to the decoder that has a similar architecture: it is made of an "usual" Transformer decoder and a variational MDN that is sampled to obtain a prediction that will be combined to the Transformer output by (again) a gating mechanism.

The model can be trained in an end-to-end fashion using the "reparametrization trick" of Kingma and Welling (2014). In order to ensure that the estimated variances for the variational posteriors are positive, we used the *softplus* activation function (Zheng et al., 2015), as done in van den Berg et al. (2018)'s implementation.

In addition, concerning the training of the decoder's MDN, we compare two different ways to compute the mixing coefficient: the first one consists in a vanilla non-latent $\mathrm{softmax}$, the second on a relaxed categorical variational method that relies on a Gumbel-Softmax sampling (§2).

The model has been implemented in `OpenNMT-py`[3] (Klein et al., 2018).

**Encoder** Our encoder backbone is the "standard" transformer of Vaswani et al. (2017), made of 6-layered transformer layers each with 8 attention heads. the *feed-forward* layers have 2,048 parameters and the dimension of lexical embeddings is 512. The dimension of the encoder vari-

ational network is 128. The network is extented with 4-flows Normalizing Planar Flows (Rezende and Mohamed, 2015).[4]

Following to Setiawan et al. (2020), we combine the last Transformer layer output to the latent vectors using a gating mechanism. We used a feed-forward network to transform the representation of dimension 128 predicted by the variational network into a representation of size 512 that matches the Transformer representation dimension.

In Figure 1b, we show the Transformer and variational encoding latent state $\boldsymbol{z}$ as being estimated ($p_\theta(\boldsymbol{z}|\boldsymbol{x})$) approximating the posterior's mean and variance, both learned using the reparametrization trick. In the figure, we can also observe how our model's encoder comprises the Transformer backbone and VI network.

**Decoder** As for the encoder, the first component of the decoder is the "standard" Transformer decoder of Vaswani et al. (2017) and uses the same parameters as the Transformer encoder.

The Transformer decoder's last layer output is passed to a 128-component MDN, with trainable parameters $\phi$ and $\pi$: $\phi$ encodes the mean and variance of each multivariate gaussian components; $\pi$ contains the probabilities of the categorical distribution that generates the mixing coefficient for each component. Concisely, we estimate a posterior as a series of $M$ posteriors parameterized by $\langle \phi, \pi \rangle$, i.e. $\tilde{q}_m^{\phi;\pi}(z_{dec}|\boldsymbol{x}, \boldsymbol{y}_{1:t-1})$, conditioned via the decoder's Transformer, on both the gated latent encoder's output and previous pre-

---

[3] https://github.com/josecar25/MDN-VNMT

[4] We used the implementation of https://github.com/riannevdberg/sylvester-flows

| Corpus | #sentences | #tokens | ASL | TTR | #chars |
|--------|-----------|---------|------|------|--------|
| *train set* | | | | | |
| WMT | 2.2M | 64.2M | 29.7 | 0.20 | 335 |
| OpenSub. | 9.2M | 57.7M | 6.73 | 0.18 | 428 |
| | | | | | |
| *test set* | | | | | |
| OpenSub. | 11,000 | 66,148 | 6.01 | 0.23 | 111 |
| newstest | 3,003 | 68,155 | 22.70 | 0.23 | 111 |

| Corpus | #sents | #tokens | ASL | TTR | #chars |
|--------|--------|---------|------|------|--------|
| *UGC test* | | | | | |
| PFSMB | 777 | 13,680 | 17.60 | 0.32 | 116 |
| MTNT | 1,022 | 20,169 | 19.70 | 0.34 | 122 |
| | | | | | |
| *UGC blind* | | | | | |
| PFSMB | 777 | 12,808 | 16.48 | 0.37 | 119 |
| MTNT | 599 | 8,176 | 13.62 | 0.38 | 127 |
| 4Square | 1,838 | 18,234 | 9.92 | 0.22 | 109 |

Table 1: Statistics on the French side of the corpora used in our experiments. TTR stands for Type-to-Token Ratio, ASL for average sentence length and #chars for the number of different characters.

dicted tokens, $y_{1:t-1}$. The MDN's mixing coefficient ($\alpha_m(x, y_{1:t-1})$) network also takes the same input and is computed separately, by either using a fully-forward layer with softmax activation or the relaxed categorical Gumbel distribution. Both networks computing $\tilde{q}_m$ and $\alpha_m$ are jointly trained in an end-to-end fashion, such that translation loss is minimal for representations sampled from the resulting mixture, obtained according to Equation 4.

## 4 Training models

All systems are trained using a batch size of 4,096 tokens using the Adam optimizer (Kingma and Ba, 2015) accumulating gradients every 2 steps, and the Noam learning rate schedule (Vaswani et al., 2017) with 8K warmup steps. Throughout training, learning rate attains a maximum of $7e-4$ and minimum of $1e-5$. Both encoder and decoder Transformers are trained using 0.1 dropout and we employed 0.1 label smoothing (Szegedy et al., 2016). Training for, at most, 300K training iterations on a single Nvidia V100 took about 40 hours to converge for the multi-VNMT models, 34 hours for VNMT-baseline and 28 hours for the non-latent Transformer baseline. In order to avoid posterior collapse, and as done in Setiawan et al. (2020), we use $\beta_C$-VAE (Prokhorov et al., 2019), with values $\beta = 1$ and $C = 0.1$. Additionally, we used a Kullback-Leibler (KL) annealing schedule of 100K iterations for training. We set a 10% probability of dropping the target word (Bowman et al., 2016). We have chosen, as initial experimental configuration, $\tau = 1.0$ for the Gumbel-Softmax sampling temperature, which was selected mainly aiming to avoid artificial gradient scaling during backpropagation (c.f. Equation (5)). A beam of width 5 has been used for evaluation.

## 5 Experiments

**Datasets** We train our different MT models on two different French to English canonical parallel corpora: the first one is a subset of the WMT corpus, i.e. Europarl (v7) and NewsCommentary(v10) (Bojar et al., 2015) and the second one is theOpenSubtitles'18 corpus (Lison et al., 2018). We used BPE tokenization (Sennrich et al., 2016) with 16K merge operations.

Detailed statistics on our corpora can be found in Table 1.

**UGC Test Sets** To evaluate the different NMT models, we consider two data sets of manually translated UGC: MTNT (Michel and Neubig, 2018) and the Parallel French Social Media Bank corpus (PFSMB) (Rosales Núñez et al., 2019)[5] which extends the French Social Media Bank (Seddah et al., 2012) with English translations. These two data sets raise many challenges for MT systems: they notably contain characters that have not been seen in the training data (e.g. emojis), rare character sequences (e.g. inconsistent casing or usernames) as well as many OOVs denoting URL, mentions, hashtags or more generally named entities (NE). Most of the time, sOOVs are exactly the same in the source and target sentences.

We also consider the 4Square corpus (Berard et al., 2019) as a blind test to validate our conclusions. To analyze our neural representations (§7), we use a subset of the PFSMB, called PMUMT, which contains 400 annotated and normalized French to English UGC sentences (Rosales Núñez et al., 2021).

**Protocols** Translation quality was evaluated using BLEU (Papineni et al., 2002) and chrF2 (Popovic, 2017) both computed by SACREBLEU

---
[5]https://gitlab.inria.fr/seddah/parallel-french-social-mediabank

(Post, 2018) with the 'intl' tokenization, after detokenizing the systems outputs.

In all the experiments we used the hyperparameters values reported by Vaswani et al. (2017) and only choose the number of components of the MDN and the dimension of the latent representation on the validation set.[6] Regarding the latent dimension, we conducted the same study with 128, 256 and 512 dimensions, with 128 being the best value. A beam of size 5 has been used for evaluation.

# 6 Results

In this section we present the main MT results to study MT performance of our methods.

**MT Performance** Our first experiment aims to compare the performance of `multi-VNMT`, the model we introduced in Section 3, to that of a "vanilla" Transformer model and of a state-of-the-art VNMT system using NF.[7] The first baseline, a non-latent NMT architecture, `Transformer`, corresponds to our model without its VI components (i.e. with only the Transformer encoder and decoder); the second baseline, `VNMT-baseline`, corresponds to the equivalent of our NF setup (featuring 4 Planar Flows) from Setiawan et al. (2020).

Results achieved by these systems are reported in Table 2. We computed the 95% statistical significance by using a 1,000-samples bootstrapping, as in Koehn (2004). It should first be noted that the performances of the three systems we consider are identical when they are evaluated on in-domain data, whatever the evaluation measure considered (no statistically significant difference between the models). This observation highlights one of the strength of the proposed method: contrary to fine-tuning (arguably the most common method to adapt a system to a new domain) that often hurts performance on in-domain evaluation because of catastrophic forgetting (McCloskey and Cohen, 1989), the improvement of the quality of UGCs by the proposed method is not at the expense of the quality of translation of canonical texts.

It also appears that, on out-of-domain text, `multi-VNMT`, the approach proposed in this

work, outperforms the standard `Transformer` model as well as the state-of-the-art VNMT model, supporting our hypothesis that considering several variational inference components allows to better capture all the variations that can be found in UGC and will result in improved translation quality. Interestingly, our system also performs better than `Transformer` when evaluated on out-domain canonical data and not only on UGC data. It should be noted, however, that the gains of our model are consistent but small and statistically significant mainly when translation quality is evaluated with `chrF2`.

**Ablation study** To better understand the impact of the different components of our model, we conduct an ablation study whose results are reported in Table 3. Overall, we obtain the best BLEU scores across all test sets for the "full" `multi-VNMT` model.

In particular, it appears that *static* latent representation (`z static` in Table 3), where instead of sampling from the learned distributions, we retrieve their mean as output, show slightly better BLEU scores when translating the `MTNT` with the model trained on `OpenSubtitles` and the `newstest'14` test set with the model trained on `WMT` (+0.1 improvement in the two cases). However, results are inconsistent for UGC test sets and otherwise worse than those of the full model for both in-domain and canonical OOD test sets for our two training configurations. This might be explained by the lack of stochastic perturbations provided by the sampling step during training, leading the model to lose generalization during evaluation.

It is also interesting to note that using a categorical variational version of the mixing coefficients rather than the usual choice of computing them with a `softmax` improves translations quality: the latter is only performing better for the `newstest'14` test set when training on the `OpenSubtitles` corpus ($\pi$ *non-latent*). Following the same trend, the `WMT` training data configuration also show improvements when using the Gumbel-Softmax version, for which +0.8 and +0.3 BLEU point improvement were obtained for both the `PFSMB` and `MTNT` UGC testsets, respectively.

**Posterior collapse** We have computed the average KL divergence of the variational decoder's block (i.e. $D_{KL}\left(q_\phi(z|x,y)||p_\theta(z|x)\right)$ on the encoder side) of `multi-VNMT` and its ablated ver-

---

[6]For the number of components we tested the following values 8, 16, 32, 64, 128 and 256 and found the optimal value to be 128.

[7]We re-implemented the system of Setiawan et al. (2020).

|  |  | WMT | | | | OpenSubtitles | | | | |
|---|---|---|---|---|---|---|---|---|---|---|
|  |  | PFSMB† | MTNT† | News◇ | OpenSubTest | PFSMB† | MTNT† | News | OpenSubTest◇ | # params. |
| BLEU | Transformer | 15.1 | 21.3 | **27.9** | 16.4 | 27.7 | 28.4 | 26.4 | 31.4 | 69M |
|  | VNMT-baseline | 15.5 | 21.4 | **27.9** | 16.4 | 28.0 | 28.9 | **26.5** | 31.4 | 72M |
|  | multi-VNMT | **16.0*** | **21.8** | 27.9 | **16.7*** | **28.4** | **29.2** | 26.4 | **31.5** | 77M |
| chrF2 | Transformer | 37.8 | 45.1 | 54.4 | 38.6 | 46.9 | 48.3 | 52.6 | 48.9 | 69M |
|  | VNMT-baseline | 38.3 | 45.1 | **54.6** | 38.6 | 47.6 | 49.2* | **53.1*** | 48.9 | 72M |
|  | multi-VNMT | **38.5*** | **45.5** | 54.6 | **39.0*** | **47.7*** | **49.6*** | 52.9* | **49.0** | 77M |

Table 2: BLEU and chrF2 test scores for our models. The † symbol indicates the UGC test sets, and ◇ in-domain test sets. Highest metric for each test set are in bold; scores significantly better than Transformer ($p < 0.05$) are marked with a *.

|  | WMT | | | | OpenSubtitles | | | | |
|---|---|---|---|---|---|---|---|---|---|
|  | PFSMB† | MTNT† | News◇ | OpenSubTest | PFSMB† | MTNT† | News | OpenSubTest◇ | # params. |
| multi-VNMT | 16.0 | **21.8** | 27.9 | **16.7** | **28.4** | 29.2 | 26.4 | **31.5** | 77M |
| $\pi$ non-latent | 15.8 | 21.0 | 27.8 | 16.4 | 28.1 | 28.5 | **26.6** | 31.3 | 77M |
| -NF | 15.3 | 21.6 | **28.0** | 16.5 | 28.3 | 28.8 | 26.1 | 31.3 | 76M |
| Z STATIC | **16.5** | 20.9 | **28.0** | 16.4 | 28.1 | **29.3** | 26.2 | 31.4 | 76M |
| -MDN | **16.5** | 20.9 | 27.8 | 16.6 | 27.7 | 28.7 | 26.2 | 31.3 | 73M |

Table 3: BLEU test scores our ablated variants. The † symbol indicates the UGC test sets, and ◇ in-domain test sets.

sion without the MDN module in an in-domain setting. When trained (using the same KL annealing schedule) on OpenSubtitles (resp. WMT) this divergence is 0.21 (resp. 0.38) for multi-VNMT and 0.15 (resp. 0.33) when removing the MDN block, suggesting that our proposed architecture is less prone to suffer from the posterior collapse phenomenon.

## 7 Analyzing Latent Representations

In this Section, we describe several experiments aiming at understanding how multi-VNMT uncovers more robust representations than the VNMT baseline.

**Impact of Noise in the Source** First, to evaluate the perturbations that the model suffers when noise is present in the source, we measure the cosine similarity between the representations of the French noisy sentences and their normalized version, taking advantage of the PMUMT corpus (§5). More precisely, we compare the source-side embeddings of the 400 original noisy UGC sentences and their corresponding 400 fully-normalized versions built by VNMT-baseline and multi-VNMT. We observe that the average cosine similarity between the noisy and normalized learning representations of multi-VNMT is 0.36 compared to an average similarity of 0.26 for the representations of VNMT-baseline, sug-

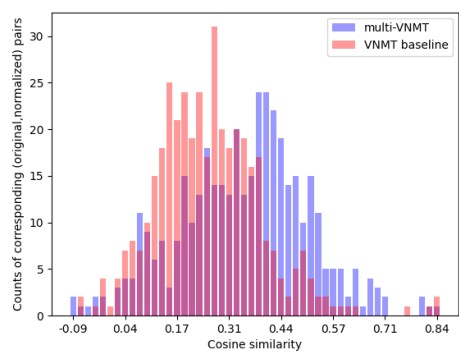

Figure 2: Distribution of cosine similarities between the representations of noisy and normalized sentences of PMUMT built by the encoder of VNMT-baseline and multi-VNMT.

gesting that the former provides more robust representations of UGC than the latter, a conclusion supported by the distribution of similarities shown in Figure 2.

**Noisy vs normalized data** To complete the previous analysis, we have reported, in Figure 3, the projection of the representations of noisy and normalized sentences computed by t-SNE. We can notice how both VNMT systems have a tendency to separate noisy and normalized sentences compared to Transformer, while both having higher cosine similarity than the latter.

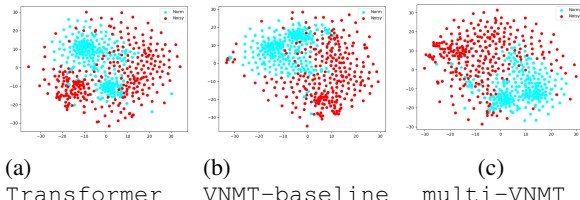

| | | |
|---|---|---|
| (a) Transformer | (b) VNMT-baseline | (c) multi-VNMT |

Figure 3: t-SNE projection of the encoder source embeddings for noisy sentences and their normalized versions.

| | PFSMB$^\dagger$ | MTNT$^\dagger$ | News | OpenSub.$^\diamond$ |
|---|---|---|---|---|
| Transformer | 27.7 | 28.4 | 26.4 | **31.4** |
| Pre-trained init. | **29.0** | 28.2 | 26.2 | 31.3 |
| Frozen embs. | 28.4 | **28.9** | **26.8** | 31.3 |
| Fine-tuned | 28.4 | **28.9** | 26.5 | **31.4** |

Table 4: Using VNMT-learned embeddings for transfer robust learned representations to the Transformer. The $\dagger$ symbol indicates the UGC test sets, and $\diamond$ in-domain test sets.

**Transferring learning representations** As discussed above, in Figure 3 we noticed that VNMT seems to enforce noisy morphology modeling to the Transformer's embeddings in an implicit way. This motivated us to study whether the information in such learning representations can be used by the Transformer backbone model and benefit from improved robustness while removing the direct latent space contribution, and notably, with the same number of parameters and architecture as Transformer. Thus, in Table 4, we report BLEU scores for the Transformer model trained on OpenSubtitles, by either initializing the VNMT-pretrained source-side embeddings before training, or fine-tuning (FT) the system. We have performed FT using the same data configuration as in OpenSubtitles and continued training for 3 epochs from the Transformer model in Table 2 while replacing the Transformer's source embeddings by their VNMT-learned version's weights.

Results in Table 4 provide evidence that VNMT enforces more robust embeddings, which perform consistently better over the PFSMB UGC test set compared to the baseline, the system Frozen embs giving the most consistent results over UGC. This system also achieves the best newstest'14 canonical OOD test set in the OpenSubtitles setup, while taking advantage of an increased robustness to UGC. These results

| | PFSMB (Blind) | MTNT (Blind) | 4Square |
|---|---|---|---|
| Transformer | 19.7 | 25.0 | 21.9 |
| +FT emb. | 19.4 | 25.3 | 22.0 |
| VNMT-baseline | **20.0** | 25.3 | 22.0 |
| multi-VNMT | **20.0** | **26.4** | **22.5** |

Table 5: BLEU scores of our best systems on blind test sets.

indicate that our VNMT model leads to embeddings that are more robust to noise even when used in a classic transformer-based NMT baseline. An interesting path of research would be to evaluate these embeddings in other tasks and scenarios (e.g. Cross lingual UGC Q&A).

# 8 Blind test sets scores

We evaluated our best performing model (multi-VNMT trained on OpenSubtitles) on the blind test sets described in § 5, translating another set of tests to assess whether our approach proves useful for generalization over different types of UGC. We have also included the 4Square corpus (Berard et al., 2019) to validate our VNMT system on other domain of UGC (restaurant reviews). We also display the results when using the VNMT-baseline baseline and the Transformer model to assess improvement of our proposed architecture. We report such results in Table 5, where we can see that, when translating our blind UGC test sets, multi-VNMT consistently outperforms the baselines. It is interesting to notice that, although the in-domain performances for these 3 systems are very similar (between 31.4 and 31.5 BLEU in Table 2), the performance gap of blind UGC test sets is larger, i.e. +0.8 BLEU in average compared to the non-latent baseline.

# 9 Discussion

**How MDN behaves under noise** In Appendix A, we discuss how MDN components are activated when translating canonical in-domain and OOD texts, as well as UGC and normalized UGC. In Figure 4 and Table 6 in the Appendix, we show that noisy UGC activates MDN's components with low correlation to other OOD canonical texts and even to its normalized version, which implies that the distribution of the kernels' mixing coefficients is relatively among, the 4 test sets

considered, unique, i.e. relatively uncorrelated from the activation of other canonical texts (indom and OOD), when processing UGC. We cannot conclude, however, whether this observation is a consequence of the noise propagated through the model's networks, but the enhanced robustness we witnessed in the translation results (much better performance to UGC, while keeping on-par or slightly better canonical (in-domain and OOD) performance) suggests that these mixing coefficients (that ultimately control the final decoding output) activate different variational posteriors (one per kernel) that can better process UGC.

**Conclusions** We introduced a novel VNMT architecture that provides improved performance and robustness over a state-of-the-art VNMT model, specifically when translating French UGC. An ablation study and blind test sets evaluation validate our architecture choice in regards of robustness capabilities for such texts. In addition, by exploring the learning representations trained by our VNMT model, and through conducting transfer learning experiments with such, we investigate the robustness brought to UGC, and show that VNMT enforces such property to the backbone model, bringing a promising avenue for more robust pre-trained neural learning representations. However, an open question arising from this work, it is currently unclear if the performance gain we observed is due to a better generalisation to distributional shift or if it corresponds to a better adaptation to noise in the input. Future works will be devoted to this question, which can be abstracted away to study whether UGC idiosyncrasies are a form of noise, some parts being learnable, or are rather points to a new domain.

## Acknowledgments

We thank our anonymous reviewers for providing insightful comments and suggestions. This work was funded by the ANR projects ParSiTi (ANR-16-CE33-0021). This work was granted access to the HPC resources of IDRIS under the allocation 2021-AD011011748R1 made by GENCI.

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

# A  How do MDN's components react to UGC?

We proceeded to analyze and visualize how the MDN mixture coefficients react when translating our different test sets. In order to do so, in Figure 4 we report results for the canonical test sets, the normalized `PMUMT` corpus, and its noisy original UGC version. Each bar of the Wind Rose diagram represents one of the 128 independent trained distributions' mixture weights, which have been normalized and scaled across the four graphics, and where the 7th MDN component seems to be consistently the one that drives most of the decoding for the presented experiments. Furthermore, we can notice that most mixing coefficients are, for the most part, have around 50% probability of contributing to the final inference mixture, despite not enforcing this behavior with any specific method (e.g. dropout). On the other hand, the visualization suggests that both yellow (50-60%) and blue components (30-40% of activation) are variable across test sets, being very similar between `PMUMT Norm` and `OpenSubTest`, which could indicate that the mixture components are learning to encode different types of texts, potentially working as an implicit topic modeling module. Regarding the visualization when translating `PMUMT Noisy`, the main MDN component identified above, seems less important even when compared to the out-of-domain `newstest'14` set, which suggests that the MDN uses more dense representations when processing noisy texts.

In parallel, in Table 6 we display the covariance of these coefficients' distributions between the combinations of their values when translating different kind of texts, along with the standard deviation and sparsity to describe how the MDN's components behave.

Comparing the visualization in Figure 6, we can notice how the noisy UGC `PMUMT` and the out-of-domain `newstest'14`, diverge from the in-domain `OpenSubTest` and normalized UGC `PMUMT` corpus. This correlation is evidenced in the results in Table 6, where `PMUMT` noisy has the lowest score when compared to every other corpus, even if its normalized version seems to

be the most correlated to the in-domain evaluation. Specifically, `PMUMT Noisy` is the least correlated to in-domain `OpenSubTest` and out-of-domain `newstest'14` corpora, which points to the MDN reacting differently to content domain and UGC specificities in the noise; this observation is also supported by the associated figure. It is also interesting to notice that, according to the standard deviation and sparsity values, the active MDN components are more dense and variable for out-of-domain evaluation conditions, for the same Gumbel sampling temperature value.

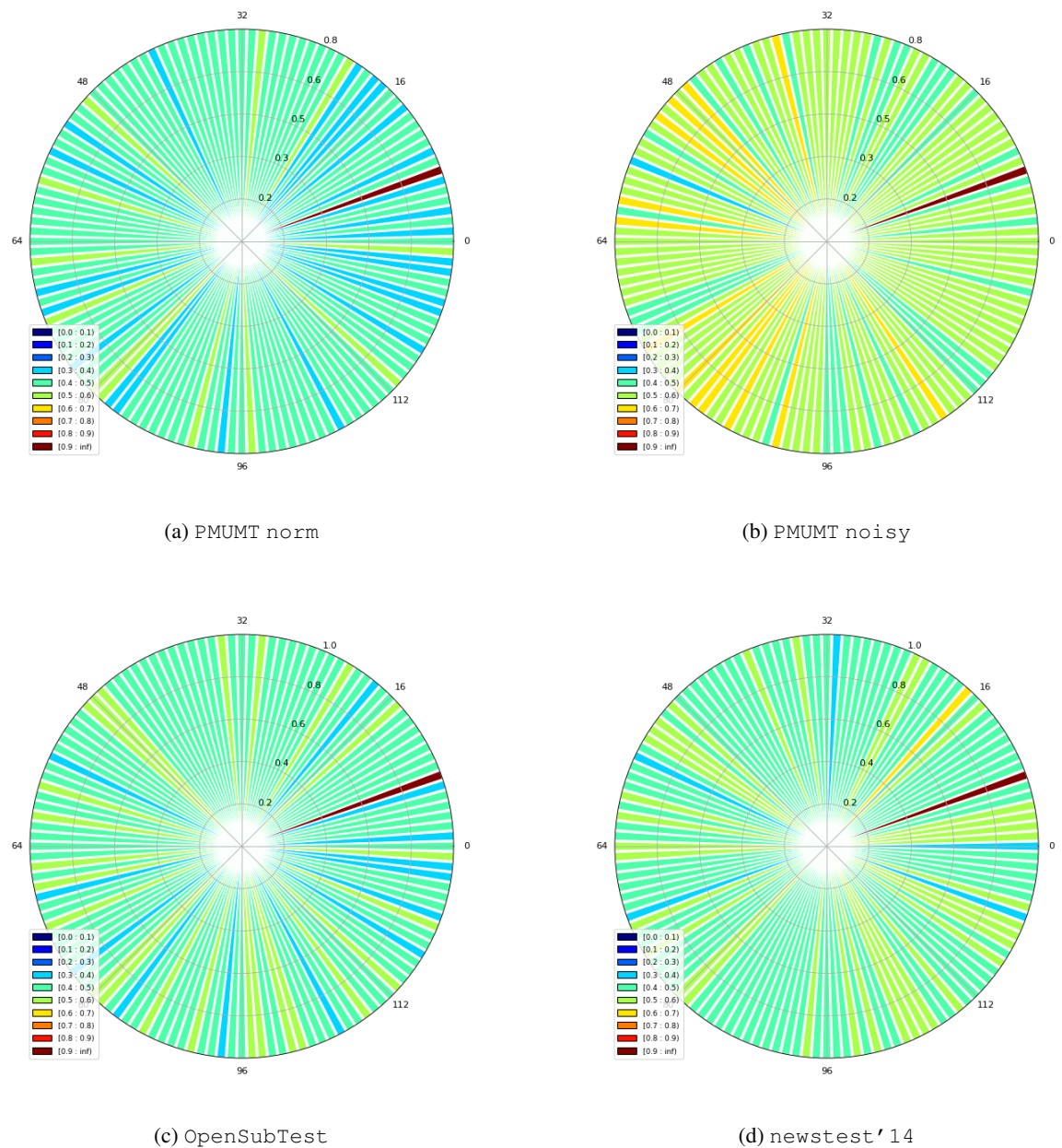

(a) PMUMT norm

(b) PMUMT noisy

(c) OpenSubTest

(d) newstest'14

Figure 4: Average MDN mixture weights for test sets of different natures.

| | PMUMT Noisy | News | OpenSubTest | std. | sparsity |
|---|---|---|---|---|---|
| PMUMT Norm | 8.16 | 9.71 | 13.05 | 1.2e-3 | 0.387 |
| PMUMT Noisy | — | 7.72 | 7.86 | 1.0e-3 | 0.382 |
| News | — | — | 9.42 | 1.1e-3 | 0.384 |
| OpenSubTest | — | — | — | 1.1e-3 | 0.387 |

Table 6: Covariance between MDN mixture coefficients during inference for different types of test sets and sparsity for each set. *std.* stands for the standard deviation.