# OpenReview forum: "Multi-way Variational NMT for UGC: Improving Robustness in Zero-shot Scenarios via Mixture Density Networks"
_NoDaLiDa/2023/Conference — NoDaLiDa 2023_

### Official Review · Reviewer_NjA7 · 2023-03-09
**-**

**Rating:** 6
**Confidence:** 2

**Review:**

It was a difficult paper to evaluate since the topic is not in my expertise area. The authors provided a thorough review of their architecture and really made an effort to analyse the results. The improvements over the baseline however seem to be only minor.

**Paper Type:**

Long paper

---

### Official Review · Reviewer_JtiD · 2023-03-13
**good work**

**Rating:** 7
**Confidence:** 2

**Review:**

The paper proposes a work on addressing noisy French user-generated content (UGC) to English with a  variational neural machine translation model. The proposed model is tested on different benchmarks, standard French to English based on Europarl, News commentary and Open Subs corpora, and tested also on two other corpora PFSMB and MTNT as noisy UGC. The model is compared with a baseline, and evaluated using the bleu score. Apart from the translation quality performance, an analysis of the latent representation is performed. The experiments show the robustness of the model for this specific task.

All in all, the paper is clear and presents a nice contribution.
Check line 687 for the repetition, and please fix the reference, line 880.


**Paper Type:**

Long paper

---

### Official Review · Reviewer_58zm · 2023-03-13
**The authors created variational NMT architecture, which slightly improves the quality, but raises the question - is it worth it?**

**Rating:** 8
**Confidence:** 5

**Review:**

General comments
The authors set out to so improve MT output robustness by creating a novel variational NMT architecture. They show their models' capabilities by translating French user-generated content. While the core idea of Variational NMT models is strong and backed with maths, it does seem like it is not worth it - mainly because some details are still missing, like training and inference time. Additionally, the usefulness of the paper is limited since it is only tested on FR to EN translations. However, if one needed to use a VNMT model, then the improvements proposed by the authors are worth looking at. I am not convinced by the claims and most likely will not turn to VNMT architecture just yet - convince me better authors :)

Improvements:
* I really want to see training and inference times compared to the baseline. This would validate the approach much more.
* Please include some basic data about datasets (train and test) in the paper, this would make reading much easier.

Questions:
1) Why did you use 16K merge operations for BPE instead of something a bit larger like 24K or 32K - does it impact the result? I feel like this would be an interesting comparison, given the core idea that VNMT gives better (more robust) input embeddings.
2) In Table.1 you show small improvements in quality with VNMT, but you also show that the number of parameters is rising a lot - this might be an unfair comparison because you could improve the baseline with an additional layer or extra training time to probably achieve slightly better results. Do you think that the model's slight improvements could simply come from added model capacity? Perhaps even a larger vocabulary would help the Transformer baseline.

Pros
* The core idea of the paper is strong and well defined, with lots of explanations and maths behind it. The strongest point of this paper.
* Figure 1 helps to understand the model quickly and is an added bonus.
* In general, the writing is good and clean, with beautiful figures/tables to compliment the text.
* I enjoyed the (limited) results on multiple domains and comparison to the baseline. The tables were cleanly formatted and a pleasure to read.
* The ablation study (table 2) was interesting and a good addition to the paper.

Cons:
* The paper uses a lot of terms/abbreviations, which sometimes make reading quite challenging.
* I am unsure about the authors' claim about "canonical texts" - what about paracrawl or other crawled data?
* The authors only cite datasets but don't show the size of the dataset, which makes the reading a bit more challenging. I feel like this is a common problem, that I have to read/search on the side a lot for this long paper - there was still room in the 8-page limit. The same thing to test datasets - PFSMB is not a "standard" dataset, so please include some information about it in the paper.
* This paper could benefit a lot from showing training and inference times because you are showcasing a new architecture - we need the whole picture. Is marginal BLEU improvement worth it, if the model is 2x slower?

Overall:
I think that this is valuable research, but perhaps the training and testing scenario is not the best to show the potential of this approach.

**Paper Type:**

Long paper

---

### Decision · Program_Chairs · 2023-03-17

Accept